

# Collaborative caching discovery management in mobile ad hoc networks environments

Hussain Alshahrani[1], Mohamed Ahmed Elfaki[1], Hamidah Ibrahim[2] and Nawfal Ali[3]

[1] Department of Computer Sciences, College of Computing and Information Technology, Shaqra University, Shaqra, Riyadh, Saudi Arabia
[2] Department of Computer Science, Faculty of Computer Science and Information Technology, Universiti Putra Malaysia (UPM), Selangar, Malaysia
[3] Faculty of Information Technology, Monash University, Melbourne, Australia

## ABSTRACT

Accessing distant data items in mobile ad hoc networks (MANETs) environments pose a huge challenge due to mobility and resource constraints. A number of research studies have been developed to enhance data accessibility and decrease the number of pending queries. A collaborative caching technique is considered as an efficient mechanism to increase data accessibility and decrease the latency which in turn reduces the pending queries. However, the processing queries that based on their classifications along with distributed indexing have not been tackled in previous works to reduce the number of pending queries and increase the number of replied queries. A collaborative caching discovery which based on service providers (CCD) is proposed in this article, which process requests depending on their status either priority or normal. This ensures that the number of pending queries is reduced with minimum cache discovery overhead. The results of the experiment reveal that the proposed strategy increased collaborative caching discovery efficiency and outperformed the cooperative and adaptive system (COACS) in terms of increasing the number of replied queries and reduction of the pending queries by 24.21 percent.

## INTRODUCTION

The emergence of mobile computing provides the opportunity to access information at anytime and anywhere. Thus, recent advances in portable computing platforms and wireless communication technologies have sparked a significant attention in mobile ad hoc networks (MANETs) among network community. This is because of the advancements of networking solution it provides. However, as mobile hosts have inherent limitations in terms of power, storage, limited battery life, and disconnection, serving queries with minimum delay is definitely of great concern. As MANETs are rapidly growing, there is a need for approaches to improve the efficiency of serving queries in MANETs. Furthermore, enhances the performance of locating the required data items with minimum delay and pending queries is required. With the use of indexing (directory), a requested node can

Corresponding author
Hussain Alshahrani,
halshahrani@su.edu.sa

acquire a needed data item from its neighbours if it knows that its neighbours have cached the data item. Due to inherent characteristics of MANETs that are asymmetric communication cost, excessive latency, limited bandwidth, dynamic topologies and resource constraints, cooperative cache management techniques designed for wired networks are not applicable to ad hoc networks (*An et al., 2004*; *Chand, Joshi & Misra, 2007b*). Hence, retrieving remote data items is a challenging task which is costly in terms of query latency (*Safa, Deriane & Artail, 2011*). Caching frequently accessed data items is an effective technique for improving performance since requests can be served from the local cache (*Radhamani & Umamaheswari, 2010*). However, caching techniques utilized in one hop mobile environment may not be appropriate to multi hop mobile environments since the data or request may need to go through multiple hops (*Chand, Joshi & Misra, 2006*; *Chand, Joshi & Misra, 2007b*). Therefore, several approaches have adopted collaborative caching strategies, which enable mobile nodes to cache and share data items that are in their local caches.

There are two reasons why collaborative caching improves the performance of mobile query service compared to simple caching in MANETs' environments. The first reason is that in collaborative caching, the requested data items can be retrieved from neighbouring nodes, instead of requesting them from the remote database server. This reduces the risk of dropping packets caused by pending queries, network congestions and link failures. The second one is that nodes can still receive the requested data items from the neighbouring nodes while network partitioning occurs (*Tian & Denko, 2007*). Nevertheless, the existing collaborative caching approaches in MANETs have drawbacks in term of achieving low hit ratio, high delay due to a number of pending queries. This is a consequence of serving queries based on hop by hop forwarding as in *Artail et al. (2008)*, *Cao, Yin & Das (2004)*, *Chand, Joshi & Misra (2007a)*, *Chand, Joshi & Misra (2007b)*, *Du & Gupta (2005)*, *Umamaheswari & Radhamani (2015)*, *Abbani & Artail (2015)*, *El Khawaga, Saleh & Ali (2016)*, *Ting & Chang (2013)*, *Larbi, Bouallouche-Medjkoune & Aissani (2018)* and *Delvadia & Dutta (2022)*, and broadcast or flooding messages as in *Denko (2007)*, *Du, Gupta & Varsamopoulos (2009)*, *Tian & Denko (2007)*, *Ting & Chang (2007)*, *Chand, Joshi & Misra (2007b)* and *Krishnan et al. (2021)*.

This article explores the collaborative cache management issues in the context of serving query by reducing pending queries to access data items efficiently. To tackle these issues, it proposed a collaborative caching discovery approach for MANETs based on the cache discovery provider (CDP) along with the service differentiation, named collaborative caching discovery management (CCD) that based on service providers. To reduce pending queries to access data items efficiently and provide better collaborative caching performance, a distributed indexing along with service differentiation is utilized to share the cached data items' information among the requested node's neighbors. In addition, each node will share the information about data items in its cache with its CH and the nodes at same cluster. Thus, the proposed research produced a greater cache hit ratio within collaborative neighbor nodes and that it turn participated in decreasing the number of pending queries in processing requests when compared to the COACS technique (*Artail et al., 2008*) (it is the "closer" related work compared to the recent works since it applied the

query directory method in serving requests which indeed increase the number of replied queries and decrease the number of pending one).

The rest of this article is organized as follows: in the next section, a survey of the recent related work is presented. Then the martial and methods section is discussed. The performance evaluation is discussed in the following section, and followed by results and discussion section. Finally, the article is concluded along with future works remarks.

## Related works

There is a number of studies focusing in the area of mobile computing particularly in collaborative caching. This is in addition to the fact that there are a number of recent research studies still investigating the current performance and future possibilities of collaborative caching technologies in MANETs to enhance queries severing (*Artail et al., 2008*; *Yin & Cao, 2005*; *Du, Gupta & Varsamopoulos, 2009*; *Safa, Deriane & Artail, 2011*). Serving queries on mobility environments is an important topic of this research. The query is mostly processed on locally at the node the initiated the query. This allowing for real-time updates while lowering the burden on database servers.

Effective cache resolution techniques were developed to resolve the cached data request which uses a split table approach (*Joy & Jacob, 2013*) and copies of data packet using replication mechanism to improve data availability and packet loss during network traffic (*Sridevi, Komarapalayam & Kamaraj, 2020*). The main objective of these approaches is to reduce latency, to lessen flooding and network traffic, and to avoid the drawback of group maintenance by having a distributed approach. Zone Cooperative (ZC) approach is proposed to consider the progress of data discovery (*Umamaheswari & Radhamani, 2015*). In this approach, the nodes at the same cluster range form a collaborative caching since a zone transmission range is formed based on a set of one hop neighbors (*Elfaki et al., 2019*). Each node consisted of a local cache to store frequency access data not only for own request satisfaction, but also for other nodes data request satisfaction that go over it. Once the data on the server side is updated, the cached copies on the nodes become invalid. If a data miss in the local cache, the node first checks the requested data items in its zone before forwarding the request to the next node that lies on a path towards server (*Elfaki et al., 2019*). However, the latency may become longer if the requested data item is missed at intermediate neighbor's nodes. Moreover, there are also a number of research studies developed a cooperative caching strategy to improve data access, data availability, and information retrieval in MANETs (*Artail et al., 2008*; *Yin & Cao, 2005*; *Du, Gupta & Varsamopoulos, 2009*; *Wang et al., 2022*) propose two strategies: CacheData and CachePath, as well as a hybrid strategy that combines the two techniques to increase the performance by utilizing both strategies meanwhile taking into consideration their drawbacks and limitations. Cache resolution and cache management are designed by *Du, Gupta & Varsamopoulos (2009)* to increase the data availability and the access efficiency. For management issues, eliminating caching replicas is applied within the collaborative range where many data diversities are accommodated. For cache resolution, two measurements are used to evaluate the performance: average latency and energy cost per-query (*Du, Gupta & Varsamopoulos, 2009*). However, flooding is a drawback of the strategy, as it adds extra

cache discovery overhead (*Kumar et al., 2010*). *El Khawaga, Saleh & Ali (2016)* developed an adaptive cooperative caching strategy (ACCS) approach to minimize pending and query delay in MANETs. The originality of this approach is to concentrate on cache replacement and prefetching policies. ACCS is build based on table driven routing strategy without additional penalties. This approach involves gathering routing information during cluster formation and then populating the routing tables accordingly, such behavior significantly minimizing the query delay. However, this approach concerning more about prefetching policies rather than cache discovery which it works with real time requests.

The scheme proposed by *Fiore, Casetti & Chiasserini (2011)* aims to consider all the information of each group zone of the MANET as a whole. Their scheme investigates both cases of nodes with large and small cache sizes. In this scheme, nodes are allowed to decide which documents to cache and for how long this document will be cached. In *Ting & Chang (2013)*, a group-based cooperative caching scheme (GCC) is achieved. Basically, this scheme is based on the concept of group caching where each mobile host and its k-hop neighbors are allowed to form a group. Each mobile host maintained a directory of cached data items. Each mobile node (MN) broadcasts message in the group to obtain the directories of its k-hop group members. However, in this technique the overhead might be generated by mobile host's requests (*Larbi, Bouallouche-Medjkoune & Aissani, 2018*).

Advance collaborative caching is developed by *Artail et al. (2008)* to address the constraints of the technique proposed by *Yin & Cao (2005)*. The initiated queries are indexing to help in locating the needed data items according to *Artail et al. (2008)*, *Lilly Sheeba & Yogesh (2011)*, *Sheeba & Yogesh (2020)*, *Lilly Sheeba & Yogesh (2017)* and *Sheeba & Yogesh (2020)* studies. In a cooperative and adaptive system (COACS) which is developed by *Artail et al. (2008)*, nodes can play one of the two roles: caching node (CN) or query directory (QD). CN is in charge of caching data items (responses to requests), whereas QD is in charge of caching requests provided by requesting mobile nodes. In the COACS methodology, there are two methods for locating data objects. First, if the query is not served locally within the system, the query goes across number of QDs before being forwarded to the database server. Second, before matching one of these QDs' nodes, the request traverses the list of QD nodes. Although COACS solves the limitations and drawbacks of the approach which described by *Yin & Cao (2005)*, an ideal approach to optimize request processing is still needed. Once the number of requests traversed inside the QDs grows in COACS (*Safa, Deriane & Artail, 2011*), and the quires failed to be served in the system, the latency and bandwidth consumption increase too. As a result, this technique is ineffective, particularly for requests that require immediate attention and response. Therefore, the majority of the previous approaches are relaying on broadcasting messages, flooding, and hop-by-hop discovery for fetching the required data items rather than increasing the cooperation at the local cache level. Furthermore, requesting data items using broadcasting messages, flooding, and hop-by-hop add penalties on the scattered bandwidths which may increase the delay for fetching the required data items and increase the number of pending for the required data items (*Elfaki et al., 2019*).

## MATERIALS & METHODS

### The proposed collaborative caching discovery management (CCD)

The proposed CCD approach was build-up based on five possible scenarios which are identified in *Elfaki et al. (2014)* and described in Fig. 1. The load balancing algorithm has taken place to increase the number of replied queries and decreased the number of pending one. This algorithm also plays an important role for serving queries since there are more than data sources possible which are described *via* the following scenarios. In CCD, a requested node (RN) submits a query based on two classifications either priority or normal, as determined by RN. The first scenario occurs when the data item is not locally available at RN's cache. The RN checks its recent priority table (RPT) and retrieves the required data if it is available within neighbouring nodes. The data item is retrieved from the neighbour node, which can be discovered by referring to RN's RPT. If the cached data item information is not found in the RN's RPT, the query will be sent to the cluster header (CH) if the query is priority. The second scenario occurs when the data item and its cached information are not found in the local cache and the RPT of the requested node respectively. Therefore, the RN refers to its CH and retrieves the required data item. In the third scenario, the query is sent to the CH of RN when the data item is not found in the RN's local cache and its information is not indexed in the RN's RPT as well as is not found at the RN's CH. The CH forwards the query to the node having the data item within the cluster range which is known as cluster's cache hit. When RN initiated a priority query and the required data item not cached locally or in the cluster, the CH of RN directs the query to the database server, which it turns response directly to the RN in the fourth scenario. In the fifth scenario, RN initiated a normal query where the requested data item is not found at the cluster zone. The query is sent to the database server along with the RN address and the required data item is sent to the RN that if the required data item is missed at neighbour's CH level and all CHs visited.

In MANETs, serving query is a critical issue, since it may require to travel from one end of the network to reach the source node, where the required data item is send back over the network to the RN (*Artail et al., 2008*; *Idris, Artail & Safa, 2005*). This increases the number pending queries, and average query latency. Therefore, the proposed CCD implement minimum distance packet forwarding (MDPF) algorithm which similar algorithm that is used in *Artail et al. (2008)* and *Idris, Artail & Safa (2005)*. This algorithm is basically designed to explore and determine the nearest neighbour nodes. To minimize the travelling in serving queries in MANET's environment, the proposed CCD approach classified the query either priority or normal as some queries require immediate response. As well as, nodes within same cluster share their cached data information. Thus, the CH and cache node (CN) are able to index more information of cached data within the cluster range. This facilitates in determining the destination of the query. Furthermore, a list of visited destination is maintained with the traversed packet to avoid misdirection a query to a node which has been visited already in case if the query is normal. This also plays an important role for minimizing the number of pending queries and increasing the number of replying queries with less average delay. This is because of increasing the level of collaboration at

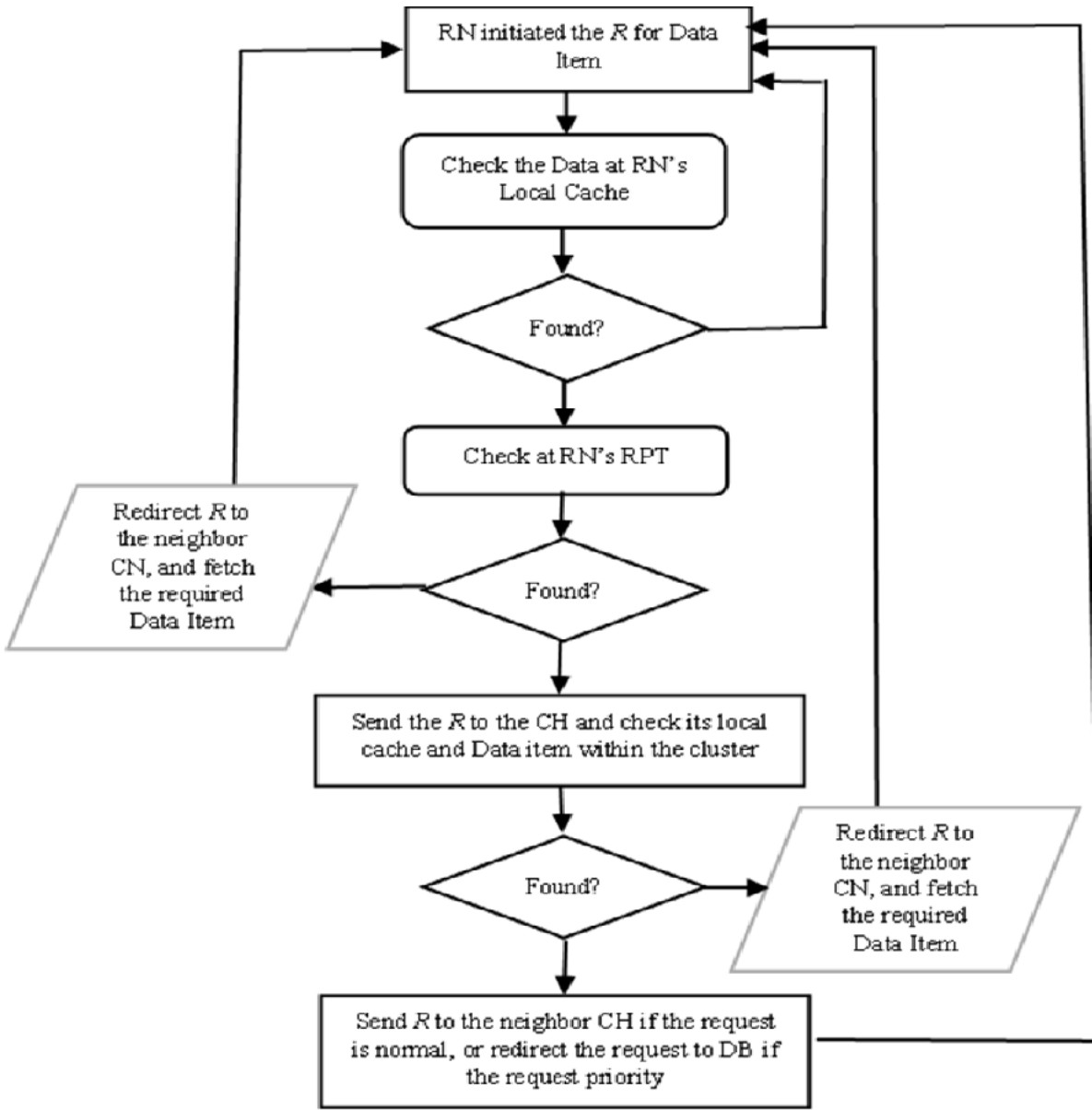

**Figure 1** Processing a request and data item replied.

the local cache among the neighbour nodes that are located in the same cluster zone. The mobile nodes are allocated geographically adjacent into the same cluster according to some rules that are distance and similar interests. This research did not develop a new cluster algorithm but applied the existing cluster algorithms as in *Chatterjee, Das & Turgut (2002)*, *Younis & Fahmy (2004)* and *Yu & Chong (2005)*.

## CCD system model

The CCD simulation model setup contains a number of clusters, C = {c1, c2, …, cn} and every cluster has a CH, a database server where each CH has a direct link to it *via* an

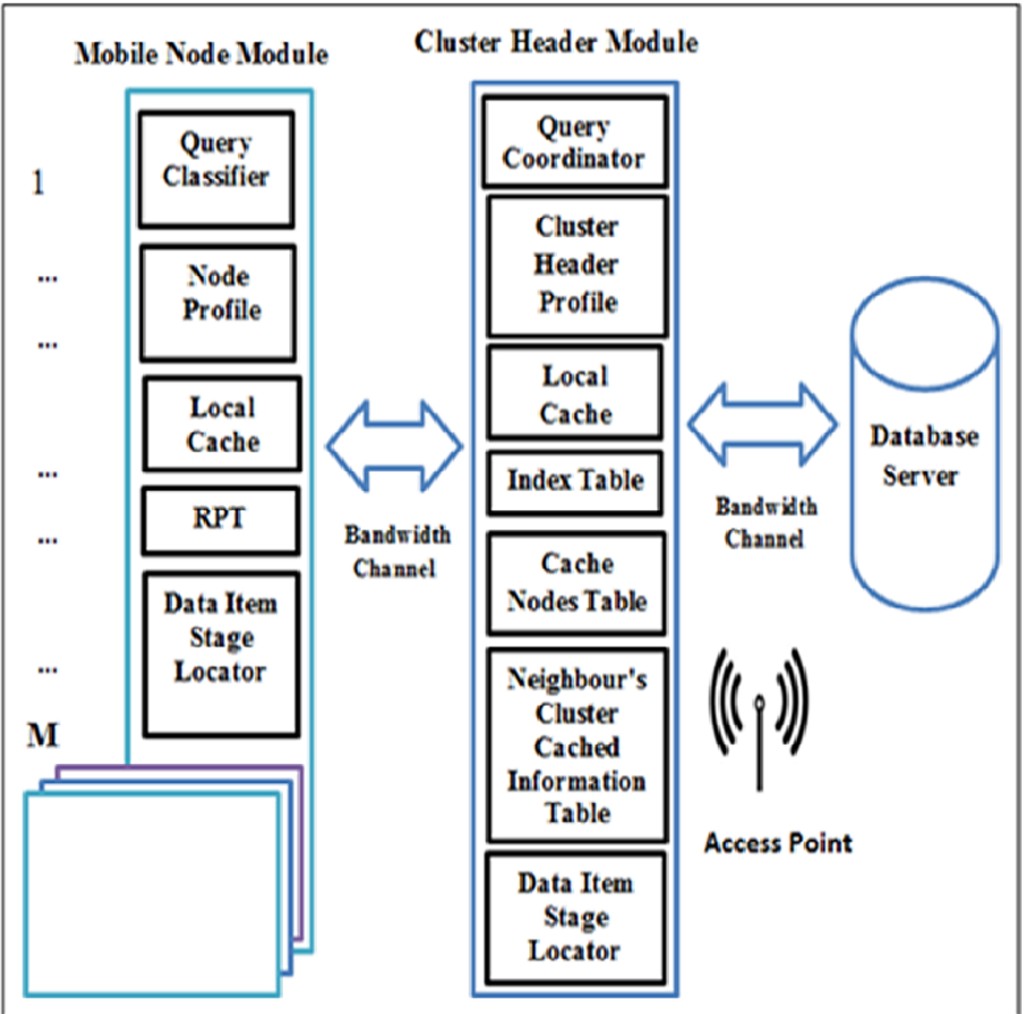

**Figure 2** System model of the CCD approach.

access point, and a number of mobile nodes, MN = {MN1, MN2, ..., MNm} where some nodes act as requested nodes (RNs) or cache nodes (CNs), while others act as RN and CN simultaneously. There are two main modules in the simulation model, namely: MN module and CH module as illustrated in Fig. 2. In MN, there are five subcomponents which are: (1) query classifier. This subcomponent is responsible to differentiate the service of a requested node's query based on the query classification. Hence, queries are coordinated to a particular data source. (2) Node profile stores the node address which is used as an identifier of a node, node type, and node capacity. (3) Local cache, which is used to cache the node owns data items. (4) RPT, which is responsible to record the information of the cached data item for neighbouring nodes. (5) Data item stage locater, which is used to discover the required data items at a particular data source.

On the other hand, the CH module consists of: (1) query coordinator, which is responsible for coordinating and controlling the forwarding requests based on their classifications. (2)

Cluster header profile, stores the cluster header address which is used as an identifier of a cluster header and its capacity. (3) Local cache, which is used to cache the cluster header owns data items. (4) Index table, which is used to track nodes entering and leaving a cluster range for updating purposes. (5) Neighbour's cluster cached information table, which records the neighbour clusters cached data item information to determine the next hop to forward queries. (6) Cache nodes table, which is utilized to allow CH to index nodes' addresses and their cached data item information within a cluster range. (7) Data item stage locater, which is used to locate the required data items *via* cluster header.

## Cluster header

The cluster header is a reasonable of coordinating nodes within the same cluster range, which is known as intra-cluster coordination. The cluster header is also in charge of communicating with other clusters and the external network on behalf of the cluster members. Furthermore, the cluster header is in charge of deciding whether to direct and coordinate queries to a specific data source, particularly if the requested data item is missing within the cluster. The decision to direct queries to a specific data source is dependent on the query classification, which is priority or normal, as well as the availability of the requested data item at the local cache or the availability of cached data item information at RPT and CH (*Elfaki et al., 2014*). Furthermore, by include an index in each node and cluster header, information is shared across mobile nodes within the same cluster as well as between clusters. This can reduce cache discovery overheads, decrease pending requests, and increase the replied one as well as the level of collaboration among neighbor nodes.

## Cluster member

A cluster member is a regular node that is not a cluster header node. Each mobile node broadcasts a hello message to other nodes in the same cluster, including its ID and cached data item metadata. Each mobile node collects topology information for its cluster using this hello message. In the proposed approach, each mobile node has a local cache area and an RPT for indexing the cached data item information of its neighbors, and it is directly connected to the cluster header (*Elfaki et al., 2014*).

## Database server

A database server is a location where original data items are recorded and organized. The database server is implemented as a fixed node in the proposed framework. Cluster headers able to access the database server *via* an access point (*Elfaki et al., 2014*).

## Wireless connection

The wireless component is used in the module to serve as a communication link between mobile nodes and the entire network. As a result, mobile nodes connect with one another over a variety of wireless channels (*Elfaki et al., 2014*).

## Data item stage locater

The stage of locating a required data items in the proposed approach is illustrated in Fig. 3. The required data item is checked first at the node local cache. In case the required data item is missed locally, the RPT is checked for required data item information. If a match

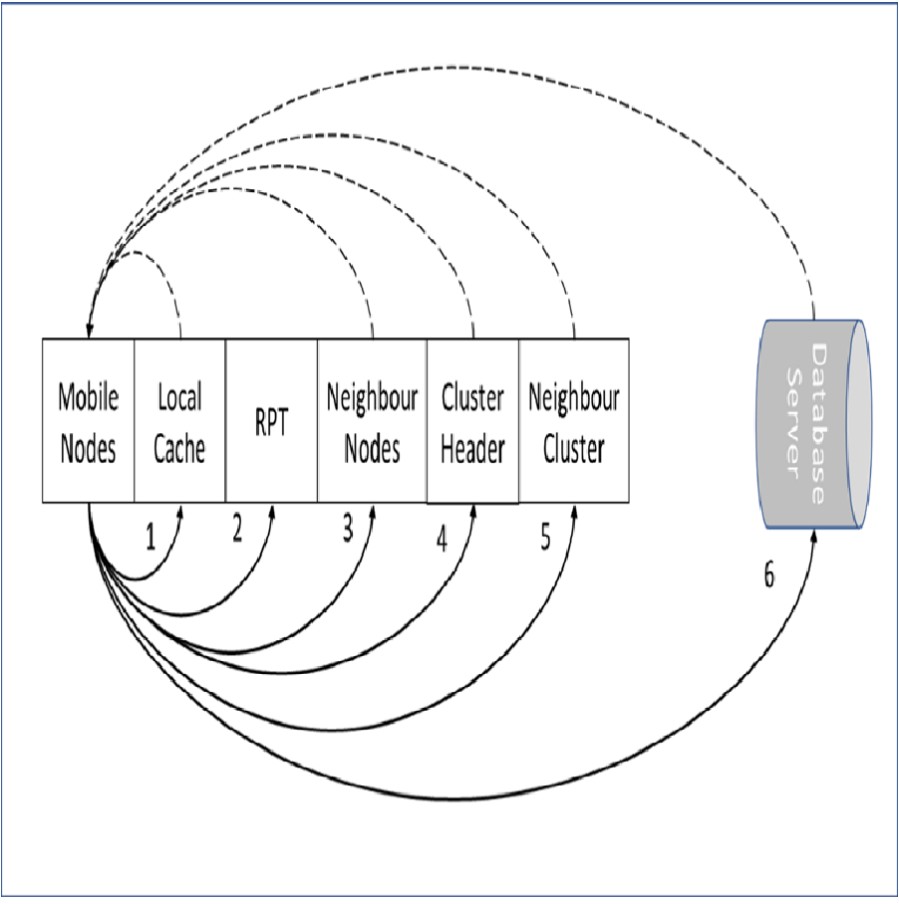

**Figure 3** Data item stage locater.

is found at the RPT, the query is forwarded to the particular neighbor's node. When the required data item is missed locally and also at the RPT, the query is directed to the CH. The CH coordinates and redirects the query to an exact data source according to the query classification either normal or priority. This is performed when the request is missed at the CH's cache. The data source can be a mobile node within the cluster zone, CH of the requested mobile node, a mobile node in the neighbor cluster or a database server.

Figure 4 shows an example of a cached data discovery in CCD. In this figure, nodes are grouped in cluster based on the same interest and distance. As mentioned in the previous section, each node cached its own data items along with the indexed data items information of its neighbor nodes at the RPT. In Fig. 4, N1 holds data items D2 and D3; N2 requests for D2 and D2 is not available at its cache, but N2 knows that N1 has D2 since N1 and N2 have shared their cached data item information earlier during the neighborhood formation. Since D2 is cached by N1, which is one (1) hop distance from N2, then N2 stores a copy of D2 from N1. If N6 requests a normal request for D7, which is located in a different cluster, the cluster header CH2 will request the D7 from its neighbor cluster.

The decision of classifying a query as priority or normal is specified by the requested node which is N6 in the example. The CH2 forwarded the N6's query to its neighbor

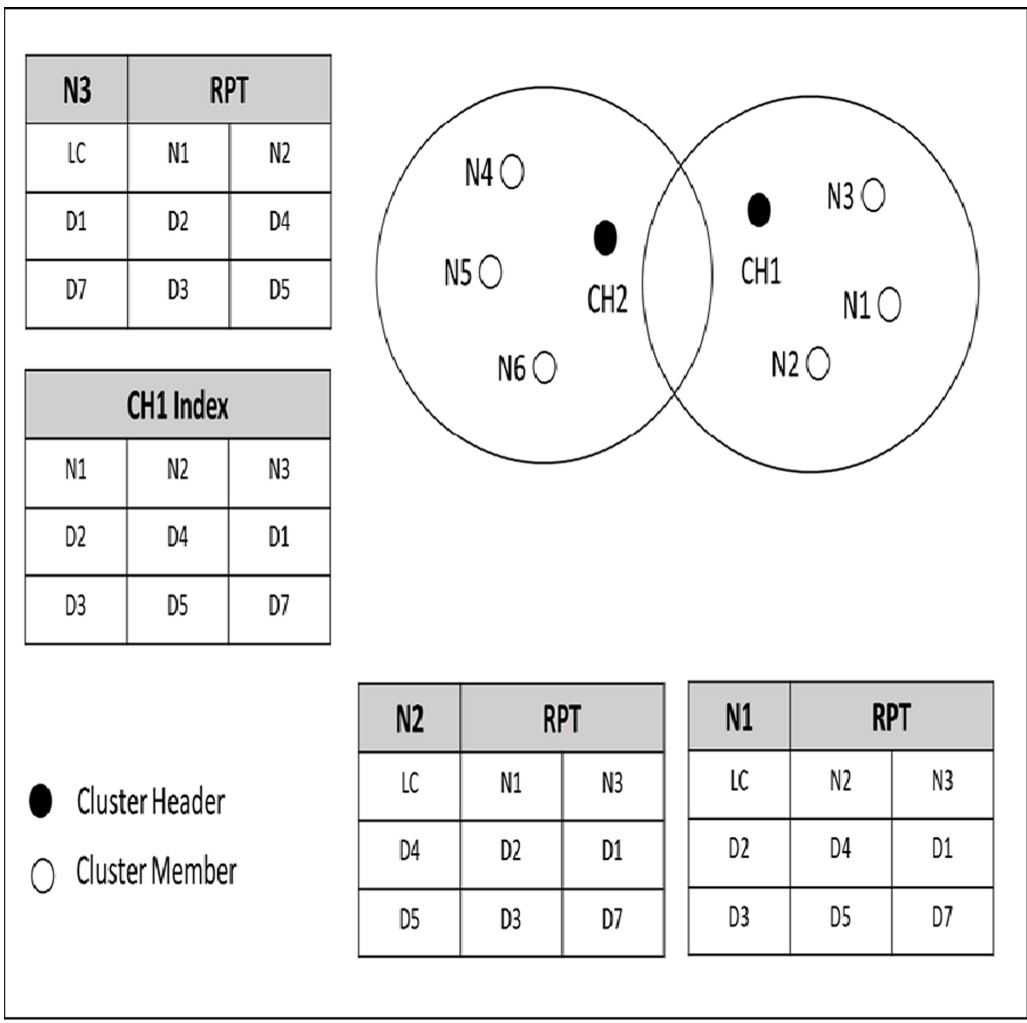

**Figure 4  Priority and normal queries.**

cluster header which is CH1. The CH1 checks its index for the corresponding data item for N6's query. If the required data item is found, the CH1 forwards the query along with N6 address to the node that cached the corresponding data item which is N3 in the provided example. On the other hand, if the query is priority and the required data item is not found within the cluster of the requested node say N5, the CH2 redirects the query to the database server to serve the query to avoid any delay. Furthermore, by enabling a query to be served according to its classification, cache discovery overhead will be reduced, since a query is guided to a particular data source.

## The load balancing algorithm in serving queries

To reduce the number of pending queries and increase the number of answering queries, a load balancing algorithm is developed to distribute sending queries across multiple data sources. Since there are more than one data sources to serve queries with the required data

items, an objective would be to reduce the number of pending queries that is handled by each node without affecting the performance of the proposed approach. In the proposed approach, queries are initiated randomly and forwarded to multiple data sources. These queries are initiated by any node in the network, which is called RN. An RN has a list of neighbour nodes to forward queries. These neighbour nodes are located one hop farther from the RN. At the cluster header level, MDPF is used to discover the nearest CH to serve the query. Tables 1 and 2 provide the symbols and the functions, respectively which are used in the load balancing algorithm that is presented in algorithm 1. In the figure, each neighbour node $i$ for a particular RN is scan (line 2). This is followed by checking the information of each data item $d$ that is cached in a neighbour node $i$ (line 3). If the information of the data item d that is listed in the neighbour node $i$ matched with the information of the initiated query $R$ (line 4), the neighbor node $i$ is added to the list of the neighbor nodes, that having the required data item (line 5). This is followed by incrementing the number of neighbour nodes that having the required data item (line 6). Furthermore, once the neighbour nodes having the required data item are determined (line 10), a neighbour node is randomly selected to serve the query R (line 11), using division and counting functions. Once the selection is made, the query R is sent to the selected neighbour node to serve the query R with the data item d (line 12), and the process is ended (line 13).

**Algorithm 1 Load Balancing**

Step 1: Countprio = 0 // initialize the number of neighbours having the required data item to zero

Step 2: *For each* neighbour *i* in RN's NB list do *// scan the current neighbours node (NB)*
Step 3: *For each* data item information *d* in neighbour *i* do *// scan the list of data items information*

                                      *// for a particular* neighbour
Step 4: *If* (*R*.Item = *d*) Then
Step 5: NB_index [Countprio] = *i // add it to the list of neighbours*
Step 6:          Countprio = Countprio+1; *// increment the number of neighbour nodes*
Step 7:     *Endif*
Step 8:     *EndFor*
Step 9: *EndFor*
Step 10: *If* (Countprio > 0) Then *// if there are neighbours having the required data item*
Step 11:    Randneight = rand() div Countprio *// randomly select a neighbour*
Step 12:    SendQuery(neighbours[NB_index[randneight]]) *// send the query to the* selected neighbour

Step 13: *EndIf*

**Table 1 Symbols of the load balancing algorithm.**

| Symbols | Description |
|---------|-------------|
| *i* | Neighbour node *i* for a particular node |
| *R.Item* | The indexed information of the requested data item |
| *R* | Request |
| div | Division function |
| Countprio | Initialize a variable for counting the number of neighbours that have the required data item |
| NB_index | Index table for recording the information of the neighbour nodes that have the required data item |

**Table 2 Functions of the load balancing algorithm.**

| Function | Description |
|----------|-------------|
| rand() | A function returns a random value between 0..1 |
| Randneight | Randomly select a neighbour node |
| SendQuery(neighbours[NB_index[randneight]]) | Send the query to the selected neighbour node |

## Performance evaluation

The NS-2 simulator software along with the Carnegie Mellon University (CMU) wireless extension (https://www.isi.edu/nsnam) was used to implement the proposed CCD and COACS (*Artail et al., 2008*) approaches. The destination sequenced distance vector (DSDV) is used as the primary routing protocol. For wireless bandwidth and transmission range, 2 Mbps and 100 m, are used respectively in this study. The mobile nodes are distributed randomly following the random way point model (RWP) movement. Moreover, the proposed CCD approach's simulation implementation area is 1,000 m ×1,000 m, and the link to the external source is established *via* the assess point (AP). The simulation setting is established in the way nodes initially are randomly dispersed, whereby each node has a random destination that moves at a random speed towards the data sources locations. The nodes speeds are set to 0.01 m/s and 2.00 m/s for lowest and highest respectively, and the pause time is set to 100s in the simulation configuration. However, as a caused of network high mobility, this study shows a scenario with a maximum velocity of 20 m/s and an average velocity of 13.8 m/s. The data source link's latency is set to 40 m/s, which is a relatively low speed according to Curran and Duffy's criteria (*Artail et al., 2008*). Table 3 lists the remaining simulation parameters, along with their corresponding values. These are the same parameters as are used in COAC approach.

The simulation square is divided into 25 clusters, each measuring 200 m ×200 m. The number of clusters is dynamic, which is consistent with the same QDs number setting in COACS (*Artail et al., 2008*). Zipf pattern access and offset values are used for nodes within the cluster and out of it, respectively. If a node in cluster x made a Zipf-like to required data item ID, the new ID would be (ID+ *nq* mod (x)) mod (*nq*), where nq is the database

**Table 3  Simulation setting.**

| Parameter | Value |
|---|---|
| Database size | 10000 data items |
| Request size | 512 bytes |
| Result size | 10 kb |
| Client cache capacity | 200 kb |
| Number of nodes | 100 |
| Simulation time | 2000 s |

size, ID is a unique ID for a specific data item, and x is the cluster range (*Artail et al., 2008*). This access pattern is used to ensure that nodes in the same range have similar interests, even if their access patterns are different. Each node is set to wait for a specified amount of time, which is equal to one second, before sending the same query again under the time out policy. After 10 s, a node sends a fresh query, if it has not received the needed data. Moreover, the CCD technique used applied algorithms called least Recently Used (LRU) to replaces old data items with new requested data items if there isn't enough capacity to cache the new one. Furthermore, time-to-live (TTL) is taken into account in this study to determined expired data items to be eliminated and replaced with new one.

## RESULTS AND DISCUSSION

Based on the discussions in earlier sections and the characteristics of the mobile computing, this research identifies a promising approach named CCD to reduce the number of pending queries and enhance cache discovery in MANETs. This research studies the effects of one of the existing collaborative caching approaches in MANET's environment named COACS (*Artail et al., 2008*) which is selected to evaluate the performance of the CCD. This is because; the COACS approach is the closest approach to the proposed CCD in tackling the issue of cache discovery in MANET's environment, even though there are a number of recent works proposed. Moreover, the most of recent works are served the queries based on broadcasting and flooding messages or caching the data item in intermediate nodes along the way from data sources, instead of guide the requests to particular sources. This section describes the results obtained from simulation modelling to validate the proposed approach and compare to the COACS approach. The result shows significant impact of the proposed CCD in improving the performance of collaborative caching management in terms of reducing the number of pending queries to access the data item. The comparison is done under different scenario environment, which are various zipf request patterns, mobile node movement (speed), node velocity, and pausing time. Pending queries have direct effect of collaborative caching performance in MANETs. This is because not all of the sending queries are successfully satisfied in a period of time, but there will be queries remain pending. Therefore, in order to increase the local cache hit and reduce the average delay, the number of pending queries must be reduced. Figure 5 illustrates the pending queries for the proposed CCD and COACS. The pending queries measured based on the simulation scenario where queries are initiated without specifying any distribution

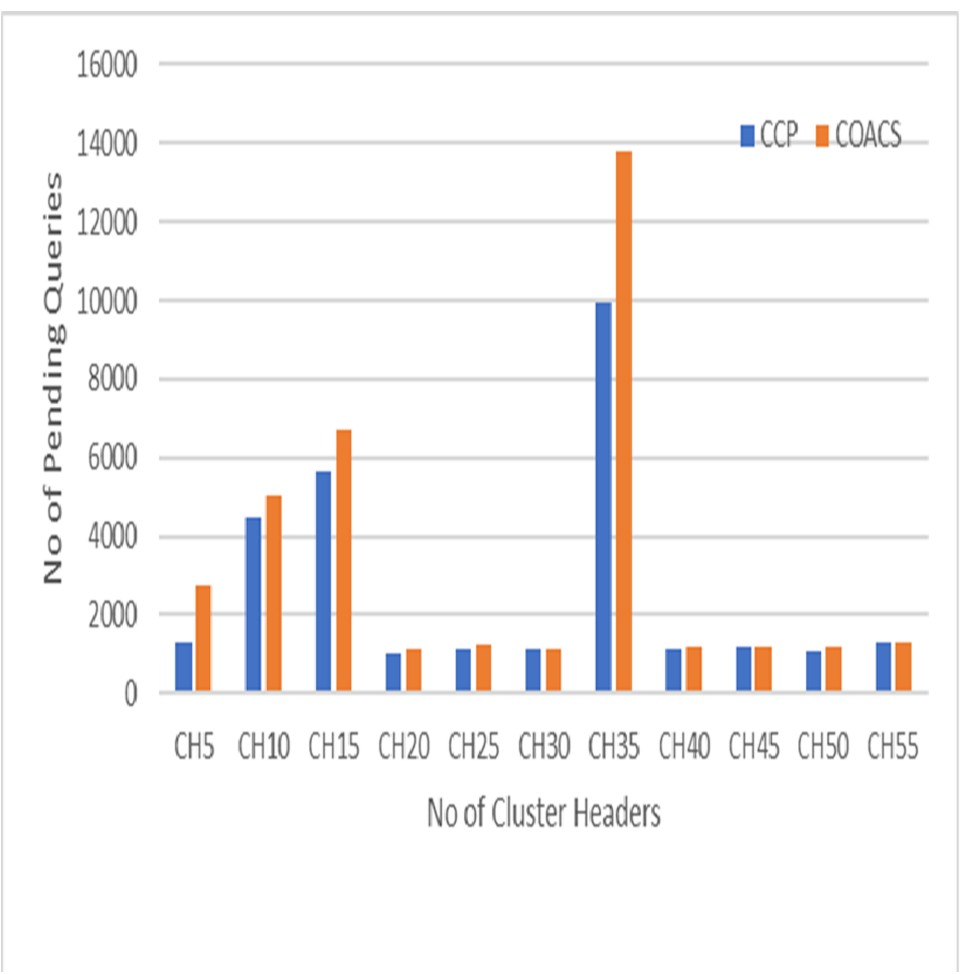

**Figure 5  Pending queries.**

of the queries types and classification. In Fig. 5, the $y$ axis shows the number of pending queries, while the x axis shows the number of CHs. It can be observed from the figure, both approaches achieved low pending queries at the beginning of the simulation. This is due to the fact that there is less number of clusters and most of the queries are satisfied within less number of forwarding hops. The number of pending queries when the number of cluster headers is five is approximately 1,800 and 2,150 for the proposed CCD and COACS, respectively. This is because at the beginning of the simulation not many queries are initiated since each node is set to initiate a query each 10 s before it can initiate a query again. As can be seen the number of pending queries increased gradually as the number of cluster headers increased. This is because the number of requested nodes increased and consequently the number of forwarding queries increased. Figure 5 also shows that the number of pending queries for both approaches is not constant as demonstrated when the number of cluster headers is 20, 21, 22, 23, ...30. The increase of the number of pending queries can be due to cluster formation caused by nodes mobility. It can also be seen in the

figure, when the number of cluster headers is 35, both approaches achieved high number of pending queries. This is due to the fact that queries are not distributed among the data sources. Furthermore, this can also be due to node mobility and cluster header and query directory reformulation for CCD and COACS. At the end of the simulation and to be exact at the cluster headers 40, 41, 42, 43, … 55, the number of pending queries decreased for both approaches. This is because neighbour nodes collaborate with each other. Furthermore, requests are served either at the local cache or inside neighbour nodes, and the majority of queries are directed to a certain data source. The results that are obtained in Fig. 5 proved that our proposed approach outperformed the COACS approach, with a decrement of 24.21% in terms of pending queries based on the same simulation scenario illustrated in section 4. The percentage of differentiation is computed using the following equation provided by *Chapra & Canale (2002)*.

$$\left[ \frac{\text{CCSP} - \text{COACS}}{\text{CCSP}} * 100 \right].$$

This is because in our proposed CCD, queries are coordinated to a certain data sources. Hence, the number of pending queries is reduced. Figure 6 on the other hand, shows the performance of our proposed CCD and COACS, in terms of the number of queries that is successfully replied. In the figure, the replied queries for both approaches are fluctuating between increasing and decreasing. The number of replied queries for both approaches is rapidly increased to reach 10,500 for the proposed CCD approach and 9,900 for COACS approach when the number of cluster headers is 10. This indicates that our proposed approach has enhanced the collaborative caching for serving queries. As demonstrated also the number of replied queries is rapidly decreased for both approaches when the number of cluster headers is 20 and 35 and almost constant till the end of the simulation. The fluctuation of increasing and decreasing the number of replied queries is caused by nodes mobility, neighbourhood formulation, and the level of collaboration among neighbour nodes.

## CONCLUSIONS

This research article examined the performance of existing collaborative caching approaches in terms of number of pending queries and replied queries. Hence, it aims to reduce the number of pending queries by distributing and guiding queries to a particular data source, and consequently increase the number of replied queries. Accordingly, this is a significant indication of increasing the collaborative level among neighbour nodes, which leads to reduce the pending queries and it turns increase the number of replied quires. The proposed CCD process the queries based on their classification. Furthermore, the service differentiation for serving queries based on their classifications along with the indexing of the cache data items is the heart of the proposed CCD approach. This has a big impact on how a delay is reduced in serving queries since are directed and coordinated to a certain data source. Additionally, it also reduces the number of pending queries. The performance of the proposed CCD approach is evaluated experimentally. The results

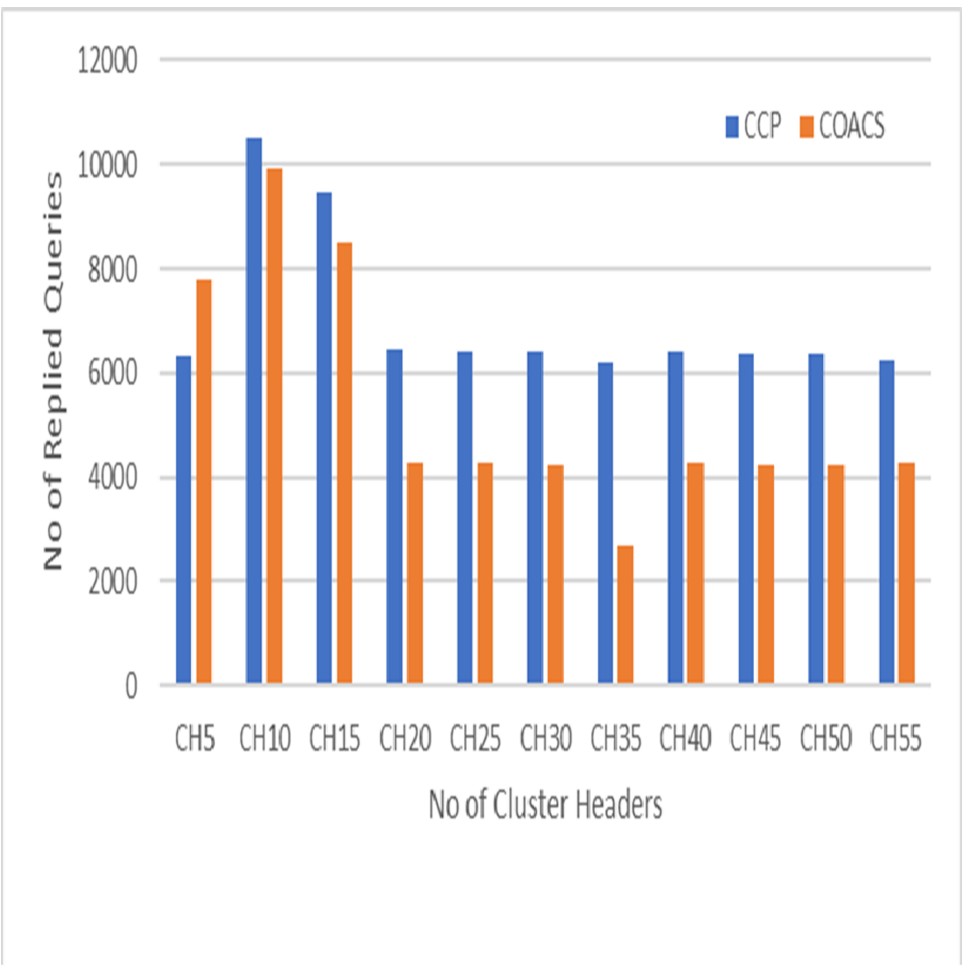

**Figure 6  Replied queries.**

reveal that the proposed CCD enhances collaborative caching efficiency and outperform the COACS, with a reduction in pending requests of 24.21 percent. In the future, a number of enhancements is needed to improve the reliability of CCD's performance and stimulate further on collaborative caching in MANET's environment, such as cache consistency, cache replication, cache pre-fetching and collaborative caching in mobile cloud computing.

## ACKNOWLEDGEMENTS

The authors would like to thank the Deanship of Scientific Research at Shaqra University for supporting this work.

### Funding

The authors received no funding for this work.

## Competing Interests

The authors declare there are no competing interests.

## Author Contributions

- Hussain Alshahrani conceived and designed the experiments, performed the experiments, analyzed the data, performed the computation work, prepared figures and/or tables, authored or reviewed drafts of the article, and approved the final draft.
- Mohamed Ahmed Elfaki conceived and designed the experiments, performed the experiments, analyzed the data, performed the computation work, prepared figures and/or tables, authored or reviewed drafts of the article, and approved the final draft.
- Hamidah Ibrahim conceived and designed the experiments, analyzed the data, authored or reviewed drafts of the article, and approved the final draft.
- Nawfal Ali conceived and designed the experiments, performed the experiments, analyzed the data, performed the computation work, authored or reviewed drafts of the article, and approved the final draft.

## Data Availability

The raw data and code are available in the Supplemental Files.

## Supplemental Information

Supplemental information for this article can be found online at http://dx.doi.org/10.7717/peerj-cs.1320#supplemental-information.

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
