# Peer review of "Collaborative caching discovery management in mobile ad hoc networks environments"

_PeerJ Computer Science, doi:10.7717/peerj-cs.1320_

## Round 0.1 · original submission · Major Revisions

Kindly address reviewers comments comprehensively.

Reviewer 1 ·

Basic reporting

1. What is the difference between the Recent Priority Table and Pending Interest Table?
2. Need careful review to remove typos.
3. Most of the related studies selected are too old. I recommend the authors update this section and provide related studies of the last 3 or 4 years.

Experimental design

4. You say the proposed mechanism provides five scenarios, why not provide algorithms for each scenario?
5. What is meant by the following sentence?
When the data item is not located in the local cache and its information is not captured in the RPT or missed at the CH, the query is sent to the cluster header in the third scenario. The CH directs a query to the node which has the data item, which is known as the cluster’s cache hit.
If a request is already reached at the CH then how can you send the received request to Cluster header?
6. Too old citations (Artail et al., 2008; Idris et al., 2005)? You need to find problems from the recent papers.
7. Why not chose NS3 instead of NS2?

Validity of the findings

8. In experiments, choose a study recently developed and compare the performance of the proposed mechanism with that of the effectiveness of the proposed mechanism.
9. Choose more metrics to evaluate the proposed mechanism.

Additional comments

The proposed mechanism needs to explain using algorithms for each scenario as the authors said the whole mechanism is divided into five scenarios.

·

Basic reporting

No comments. I have added all the review comments in the field of additional comments .

Experimental design

No comments. I have added all the review comments in the field of additional comments .

Validity of the findings

No comments. I have added all the review comments in the field of additional comments .

Additional comments

The subject is a hot topic, so I think a review of this kind of subject could be required and beneficial. However, the following adjustments should be taken into consideration:

1. Avoid the use of personal pronouns.
2. The paper topic is interesting and relevant to PeerJ journal.
3. The paper is clearly written, although it contains several language errors and needs a professional proofreading.
4. The paper is well structured.
5. At the end of the introduction, the researchers should add a few sentences describing the remaining sections of this work.
6. The authors should concentrate on what was accomplished in this review and summarize how the review was arranged.
7. Figure quality is inferior; the author needs to include high-resolution images.

Reviewer 3 ·

Basic reporting

I found the paper a little bit difficult to read, not only due to the poor grammar used throughout, but also due to the unclear structure of the argument being put across. This paper should be substantially improved by thoroughly rewriting the prose. The novelty of the proposed work should be presented in this work.
1. Abstract can be substantiated with more justification on the percentage (24.21) specified in Lines 32-35.
2. Introduction should be improved. Difference or correlation existing between
Collaborative Cache Management and Collaborative Cache Discovery can be discussed to substantiate the importance of the work.

3. Explore some recent and related work like: (many related works are from 2000-2015)

a) Enhanced Cache Sharing through Cooperative Data Cache Approach in MANET, 2020.
b) Collaborative Clustering for Cooperative Caching in Mobile Ad Hoc Networks, 2017.
c) Push-pull cache consistency mechanism for cooperative caching in mobile ad hoc environments, 2016.

4. References can be sorted based on the year of publication or the referral of the references used in the paper can be sorted.

5. Many references are with incomplete bibliographic information (like ambiguity in specifying publication details like pp. in some references and without pp. in others, for instance Lines 430-434). This must be corrected.

Experimental design

The paper can further expose in detail the contents of every sub module pertaining to both the mobile node and cluster head modules ( Fig. 1)

Validity of the findings

1. Some more results can be included, considering variations in cache sizes and data sizes.

2. Fig. 4 shows some abrupt variation when the number of cluster heads is increased to 35 with respect to Pending Queries. Similarly in Fig.5 the number of replied queries drops from nearly 10000 to 6000 when the number of cluster heads are varied from 10-15 to 20-55. Justify or introspect.

Annotated reviews are not available for download in order to protect the identity of reviewers who chose to remain anonymous.

---

## Round 0.2 · accepted · Accept

The authors have revised the paper according to the reviewers' comments, and they are satisfied with the revised version.

Reviewer 3 ·

Basic reporting

No Comment

Experimental design

No Comment

Validity of the findings

No Comment